The distribution and numbers of cheetah (Acinonyx jubatus) in southern Africa

Weise Florian J. 1 2 3
Vijay Varsha 3 4
Jacobson Andrew P. 3 4
Schoonover Rebecca F. 3 4
Groom Rosemary J. 3 5 15
Horgan Jane 3 6
Keeping Derek 7
Klein Rebecca 3 6
Marnewick Kelly 3 8
Maude Glyn 9 10
Melzheimer Jörg 11
Mills Gus 3 12
van der Merwe Vincent 3 8
van der Meer Esther 3 13
van Vuuren Rudie J. 3 14
Wachter Bettina 11
Pimm Stuart L. stuartpimm@me.com 3 4
1 CLAWS Conservancy , Worcester , MA , United States of America
2 Center for Wildlife Management, University of Pretoria , Pretoria , South Africa
3 Big Cats Initiative, National Geographic Society , Washington, D.C. , United States of America
4 Nicholas School of the Environment, Duke University , Durham , NC , United States of America
5 Range Wide Conservation Program for Cheetah and African Wild Dogs, The Zoological Society of London , London , United Kingdom
6 Cheetah Conservation Botswana , Gaborone , Botswana
7 Department of Renewable Resources, University of Alberta , Edmonton , Alberta , Canada
8 Endangered Wildlife Trust , Johannesburg , South Africa
9 Kalahari Research and Conservation , Maun , Botswana
10 Department of Conservation and Research, Denver Zoological Foundation , Denver , CO , United States of America
11 Leibniz Institute for Zoo and Wildlife Research , Berlin , Germany
12 Lewis Foundation , South Africa
13 Cheetah Conservation Project Zimbabwe , Victoria Falls , Zimbabwe
14 N/a’an ku sê Foundation , Windhoek , Namibia
15 African Wildlife Conservation Fund, Chishakwe Ranch, Savé Valley Conservancy , Zimbabwe
Roberts David
Electronic publication date: 2017 Dec 11
Publication date: 2017
Volume: 5
Electronic Location ID: e4096
Received 2017 Jul 15; Accepted 2017 Nov 6
Copyright: ©2017 Weise et al.
Copyright year: 2017
Copyright holder: Weise et al.
License: This is an open access article distributed under the terms of the Creative Commons Attribution License, which permits unrestricted use, distribution, reproduction and adaptation in any medium and for any purpose provided that it is properly attributed. For attribution, the original author(s), title, publication source (PeerJ) and either DOI or URL of the article must be cited.
License URL: https://creativecommons.org/licenses/by/4.0/

Keywords: Cheetah, Endangered species, Southern Africa, Crowd-sourcing, Distribution, Leslie Matrix model

Funding: National Science Foundation 1106401 Messerli Foundation in Switzerland Comanis Foundation Big Cats Initiative of National Geographic Society This material is based in part upon work supported by the National Science Foundation (www.nsf.gov) under Grant No.1106401. This work was also supported by the Messerli Foundation in Switzerland, the Comanis Foundation, and the Big Cats Initiative of National Geographic Society. The funders had no role in study design, data collection and analysis, decision to publish, or preparation of the manuscript.

==============================
Assessing the numbers and distribution of threatened species is a central challenge in conservation, often made difficult because the species of concern are rare and elusive. For some predators, this may be compounded by their being sparsely distributed over large areas. Such is the case with the cheetah Acinonyx jubatus. The IUCN Red List process solicits comments, is democratic, transparent, widely-used, and has recently assessed the species. Here, we present additional methods to that process and provide quantitative approaches that may afford greater detail and a benchmark against which to compare future assessments. The cheetah poses challenges, but also affords unique opportunities. It is photogenic, allowing the compilation of thousands of crowd-sourced data. It is also persecuted for killing livestock, enabling estimation of local population densities from the numbers persecuted. Documented instances of persecution in areas with known human and livestock density mean that these data can provide an estimate of where the species may or may not occur in areas without observational data. Compilations of extensive telemetry data coupled with nearly 20,000 additional observations from 39 sources show that free-ranging cheetahs were present across approximately 789,700 km2 of Namibia, Botswana, South Africa, and Zimbabwe (56%, 22%, 12% and 10% respectively) from 2010 to 2016, with an estimated adult population of 3,577 animals. We identified a further 742,800 km2 of potential cheetah habitat within the study region with low human and livestock densities, where another ∼3,250 cheetahs may occur. Unlike many previous estimates, we make the data available and provide explicit information on exactly where cheetahs occur, or are unlikely to occur. We stress the value of gathering data from public sources though these data were mostly from well-visited protected areas. There is a contiguous, transboundary population of cheetah in southern Africa, known to be the largest in the world. We suggest that this population is more threatened than believed due to the concentration of about 55% of free-ranging individuals in two ecoregions. This area overlaps with commercial farmland with high persecution risk; adult cheetahs were removed at the rate of 0.3 individuals per 100 km2 per year. Our population estimate for confirmed cheetah presence areas is 11% lower than the IUCN’s current assessment for the same region, lending additional support to the recent call for the up-listing of this species from vulnerable to endangered status.

Introduction

Assessing how many individuals of a species remain, mapping where they are, estimating declines in numbers and understanding the causes are core activities for conservation science. Although entirely familiar, these activities can pose challenges, especially for large predators that are elusive and sparsely distributed across large areas. We address these challenges for the cheetah Acinonyx jubatus in southern Africa. We notice that the International Union for Conservation of Nature (IUCN) Red List (henceforth Durant et al., 2015) has addressed these same questions for the global cheetah population in its listing and an accompanying paper (Durant et al., 2017). The Red List process solicits comments, is democratic, transparent, and widely-used. Here, we present additional methods to that process to provide quantitative approaches that may afford greater detail and a benchmark against which to compare future studies. We chose the cheetah as a case study because it affords unique opportunities and because Durant et al. (2017) recommend an up-listing of the species from “vulnerable” to “endangered” status. We aim to provide an independent process to evaluate their results that uses other approaches, new data, and alternative assessments of the data analysed.

Durant et al. (2017) estimate approximately 7,100 adult cheetahs across Africa and Asia, with five separate subspecies (Krausman & Morales, 2005). Of these, approximately 4,300 cheetahs (61%) live in southern Africa, 4,029 in our four study countries, and 2,300 cheetahs (32%) in eastern Africa. Historically, cheetahs roamed large parts of sub-Saharan Africa, but have been widely extirpated, now residing in only 22% of their historical range (Durant et al., 2017). This reflects an on-going declining population and that >75% of the species’ range exists outside protected areas where cheetahs may be exposed to high levels of threat from human persecution.

Several aspects of cheetah biology make appraisals challenging. Cheetahs are cryptic, occur over a variety of habitats (Sunquist & Sunquist, 2002) and at variable, though usually low, densities (Dalerum et al., 2008; Funston et al., 2010; Boast & Houser, 2012). In addition, important population parameters, such as survival rates and inter-birth intervals vary with several factors, including competing predators (Marnewick et al., 2009; Wachter et al., 2011) and degree of human persecution (Marker et al., 2003). Such factors differ across study areas, thus hampering extrapolation (Mills & Mills, 2014). Many studies have been limited to small areas and few animals (e.g., Boast et al., 2011, but see Van der Meer, 2016). The necessary population data to assess status, threats, and population trends adequately across landscapes are consequently hard to obtain. Thus, independent approaches could lead to different conclusions on how many cheetahs remain. In this situation, it behoves researchers gather verifiable information from as wide a variety of sources as possible and to be explicit about how these data are used to produce distribution and population estimates.

Fortunately, all big cats are photogenic; the cheetah particularly so. This affords an opportunity to incorporate crowd-sourced data across large areas to document the range and numbers of cheetahs. Citizen science is emerging as an important tool in cheetah monitoring (Marnewick et al., 2014; Van der Meer, 2016), complementing data derived from other research methods such as interview surveys (Stein et al., 2012), tracks-and-signs based methodologies (Keeping, 2014), Global Positioning System (GPS) collars (e.g., Weise et al., 2015; J Melzheimer, 2002–2014, unpublished data) and remote wildlife cameras (Boast et al., 2011). Simultaneously, some research programmes expand to national and regional scales, providing important landscape level information where most cheetahs reside.

The Range Wide Conservation Program (RWCP) for Cheetah and African Wild Dogs Lycaon pictus (IUCN/SSC, 2007; IUCN/SSC, 2012; IUCN/SSC, 2015) has collated much of the existing knowledge on cheetah distribution and numbers. In regional workshops, experts revised the range extent, assessed threats, estimated population sizes, and set suitable conservation strategies and priorities. For areas with little or no sampling effort, the assessment relied on expert opinions to inform the potential status of the species. Importantly, Durant et al. (2017) reviewed the IUCN status assessment protocol and suggested that additional mechanisms were required to determine the conservation status of cheetahs adequately, particularly outside protected areas. This prompts questions such as whether alternative approaches might be necessary for the cheetah and whether other methods can assist in poorly sampled regions. We have four aims:

(1) Our first aim was to provide an independent assessment from previous efforts, driven by maximum data gathering, and including a wealth of information previously not considered. We present a data-based appraisal and analyse the largest set of cheetah information collected to date. We do so over 6.4 years, a timeframe equivalent to approximately 1.3–1.4 free-ranging adult generations (see ‘Methods’). We outline the current known range of the species in southern Africa and estimate possible range while also providing an evidence-based update of population sizes using an ecoregion based approach with density estimates calibrated to habitat suitability.

(2) We assess the value of additional data gathering methods and the data themselves in delineating cheetah range and population status. We collect verifiable data from a wide array of public, private, and research sources across Botswana, Namibia, South Africa, and Zimbabwe, a contiguous region harbouring most remaining cheetahs (Durant et al., 2017).

(3) We establish a rigorous standard of data provenance. Existing range maps arise from a combination of direct observations and expert opinion, and thus incorporate extensive experience. That said, one cannot readily interrogate a location to know whether a species was observed there and, if so, when and by whom, or whether its presence was inferred. The results we present provide such provenance.

(4) Finally, in addition to estimating cheetah range, we estimate population based on persecution levels and study estimates of cheetah density. Combined with demographic and life history information of cheetahs, we produce a Leslie Matrix model to predict the densities of cheetah necessary to sustain known off-take levels.

Methods

Data sources for cheetah observations

Botswana, Namibia, South Africa, and Zimbabwe harbour the largest free-ranging populations of cheetah in the world, i.e., those whose movements are not effectively obstructed by fencing (Durant et al., 2015). This region also includes a managed cheetah meta-population (i.e., those within fenced areas) in South Africa (Purchase et al., 2007).

We gathered cheetah distribution information from a broad range of sources. We requested monitoring data such as GPS/VHF (Very High Frequency) telemetry locations, direct sightings, camera trap records, intensive spoor surveys with experienced local trackers, and presence-absence questionnaires from 97 research colleagues. We supplemented these data with information from government wildlife departments (their survey data), additional observations from RWCP’s Pan-African cheetah sightings database, verifiable records from the public and non-governmental organizations, and an extensive literature survey. We also included cheetah records from commercial and communal conservancies managed for tourism, wildlife or livestock purposes, hunter and farmer associations, as well as amateur, semi-professional and professional wildlife photographers.

The collection of crowd-sourced data for the survey period entailed an intensive search in English, German, and Afrikaans of online image and video repositories, social media sites, and different citizen science mapping efforts. We made every attempt to verify these sightings using the methodology outlined in Appendix S1. Finally, we consulted the scientific and other literatures on cheetahs in southern Africa and geo-referenced published information for which we had no access to original data. Again, we searched publications in English, German, and Afrikaans. We conducted literature searches in the Web of Science, the IUCN Cat Specialist Group Library and Google Scholar using “cheetah” and “Acinonyx jubatus” as search terms. We classified “research data” as original and processed records sourced from the environmental research community (either as raw or published data). “Crowd-sourced data” mean cheetah observations supplied by the public.

Our cheetah records span from 1 January 2010 to 30 April 2016, giving a survey period of 2,312 days, or 6.4 years. This timeframe reflects between 1.3 and 1.4 adult cheetah generations (Durant et al., 2015; Appendix S2: mean adult lifespan on Namibian farmlands = 4.6 years, SD = 1.8, n = 161). We received cheetah monitoring data from 39 sources. They included 66 distinct data sets and studies that ranged from local to national scale. Data included records from >30 independent camera trap surveys (often across multiple years), 10 spoor survey programmes (including multiple sites and years), nine farmland studies across the four countries, summarised positional data (mainly in the form of 10 km x 10 km resolution presence grids) from >2.7 million GPS- and VHF-telemetry locations representing 208 free-ranging collared cheetahs, and communal conservancy monitoring data. In addition, we geo-referenced published cheetah information of four predator research programmes for which we had no access to the original data. We supplemented research data with verifiable crowd-sourced data (e.g., blogs, news media, social media, citizen science platforms and wildlife photographers). Of all direct point observations (n = 19,527), more than 90% had exact latitude and longitude information, while we geo-referenced the remaining 1,832 observations to the nearest verifiable locations, i.e., a known water hole or road junction (Appendix S3). The exact location data (including GPS coordinates, date, observer, source of record, number of individuals, and type of observation) are stored on Dryad, subject to sensitivity caveats. We discarded >25,000 possible public cheetah records that could not be verified for lack of reliable time, location, and/or species evidence.

Data sources for other variables

We obtained human population data from the 2015 LandscanTM High Resolution Global Population Data Set (Bright, Rose & Urban, 2015) and livestock density data for cattle Bos taurus, sheep Ovis aries, and goats Capra hircus from the 2010 Gridded Livestock of the World (GLW) v2.01 (Robinson et al., 2014). Both datasets have approximately 1 km spatial resolution which we up-scaled to 10 km grid to match the spatial resolution of our analysis.

These data represent the best current estimates available across the study area. Botswana conducted a countrywide aerial survey in 2013 that estimated livestock densities with considerably more detail than the GLW source (DWNP, 2013). However, we did not use these data as the survey combined sheep and goats, and equivalent data sources were unavailable for other countries in our study area.

We used the Ecoregions 2017 dataset (Dinerstein et al., 2017) to describe distinct habitats and define terrestrial biomes within cheetah range. We obtained data on protected areas, including information on IUCN status, from the World Database on Protected Areas (WDPA) (IUCN & UNEP, 2016).

Distribution mapping

Cheetah presence data were collected as point or polygon data and converted to raster with 10 x 10 km spatial resolution. A pixel size of 100 km2 balances the need to protect the exact GPS coordinates of sensitive data, and its edges are only marginally longer than the average daily distance moved by a female cheetah (Wilson et al., 2013). One of the smallest published cheetah home range estimates was 126 km2 for a coalition of three males in Kruger National Park (Broomhall, Mills & Toit, 2003). Assuming these cheetahs were observed in the very centre of a 100 km2 presence pixel, their home range would extend into adjacent pixels. Therefore, we classified all pixels adjacent to observed free-range cheetah presence as likely presence for a conservative estimate of cheetah distribution.

To produce a maximum distribution estimate we determined areas with possible cheetah presence. We began with the single assumption that cheetah occur within the historical range, everywhere in the study area except for Etosha Pan in Namibia (IUCN/SSC, 2015) Beginning with areas without recent cheetah observations, we employed a three-step process for determining potential cheetah range. First, we selected a threshold of human and livestock densities above which cheetah were unlikely to reside. Second, we removed ecoregions considered inhospitable to resident cheetah populations. Finally, we used spatial clustering and adjacency to remove small, isolated patches of potential habitat. Zimbabwe was the only exception to this process due to the exhaustive survey by Van der Meer (2016).

We reviewed the distribution of presence points in relation to four interrelated factors—human population density, and densities of three major livestock species: cattle, sheep, and goats. High human population density is likely to preclude resident cheetahs (Woodroffe, 2000). In both Africa and in southern Asia, wild ungulate populations decline in areas with high livestock density due to resource limitation or where landowners are hostile toward wild ungulates (Berger, Buuveibaatar & Mishra, 2013; Georgiadis, Olwero & Romañach, 2007; Madhusudan, 2004; Ogutu et al., 2009). Such decreases could limit potential densities of wild prey for the cheetah (Winterbach et al., 2015). Increased livestock density also increases the risks of conflict for the cheetah. Farmers often are intolerant of conflict and many will attempt to kill or remove cheetah after only one or two predation incidents (Weise, 2016).

We sampled human and livestock densities within all pixels with confirmed free-range cheetah presence. We then examined the distribution of these covariates (Appendix S4) to calculate thresholds of human or livestock densities at levels that included more than 85% of free-ranging cheetah presence: >25 people per km2, >10 cattle per km2, >5 sheep per km2, >5 goats per km2. Cheetah observations above this threshold may represent outliers (e.g., potentially a non-resident individual). Cheetahs are also less likely to remain undetected at high human and livestock densities.

Table 1 Cheetah density estimates across the study area in southern Africa from 2010–2016.

Country	Survey method	Study area (100 km2)	Land use	Data collection	Ecoregions	Numbers per 100 km2	Study	
Botswana	Camera trapping	2.40	Predominantly commercial ecotourism and private holiday purposes with limited farming activities	Dec. 2012– Oct. 2013	Zambezian Mopane Woodlands	0.61	1	
Botswana	Camera trapping	1.80	Mineral extraction	Oct.–Dec. 2010	Kalahari Xeric Savanna	0.51	2	
Botswana	Spoor survey—calibrated to day range and stratified by demographic group	1,096.12	Conservation, tourism, communal pastoralism, limited fenced ranching	Feb. 2011– Dec. 2015	Kalahari Xeric Savanna	0.57	3	
Botswana	Spoor survey analysed with refined Funston et al. (2010) carnivore density formula	49.00	Conservation and tourism	Nov. 2012	Kalahari Xeric Savana	0.20	4	
Botswana	Spoor survey analysed with refined Funston et al. (2010) carnivore density formula	546.45	Conservation and tourism	2014	Kalahari Xeric Savanna	0.25	5	
Botswana	Spoor survey analysed with refined Funston et al. (2010) carnivore density formula	10.60	Game ranching	2014	Kalahari Xeric Savanna	0.59	6	
Botswana	Camera trapping and tourist observations	27.00	Conservation and tourism	Oct. 2008– Jul. 2011	Zambezian Mopane Woodlands (7% floodplain habitat)	0.60	7	
Namibia	Spoor survey analysed with Funston et al. (2010) formula	57.94	Conservation with partial communal user rights	Jul. 2014	Zambezian Baikiaea Woodlands	0.19	8	
Namibia	Camera trapping with SCR modelling analysis	463.49	Mixed cattle, smallstock, game farming, hunting and tourism	2012–2016	Kalahari Xeric Savanna and Gariep Karoo	0.70	9	
Namibia	Camera trapping with SCR modelling analysis	64.45	Mixed farming and tourism	2016	Namibian Savanna Woodland, Namib Desert and Gariep Karoo	0.20	10	
South Africa (and small extension in Botswana)	Capture-recapture model on photographs	109.32	Conservation and tourism	2006–2012; subsequent monitoring	Kalahari Xeric Savanna	0.90	11	
Zimbabwe	Sighting reports collected via interviews and citizen science platform, monitoring of known individuals through photographs collected via citizen science	233.40	Predominantly hunting and tourism, some subsistence farming	2012–2015	Zambezian Baikiaea and Zambezian Mopane Woodlands	0.18	12	
Zimbabwe	Sighting reports collected via interviews and citizen science platform, monitoring of known individuals through photographs collected via citizen science	174.23	Hunting, cattle farming, tourism	2012–2015	Zambezian Mopane Woodland and Limpopo Mixed Woodland	0.51	13	
Zimbabwe	Sighting reports collected via interviews and citizen science platform, monitoring of known individuals through photographs collected via citizen science	77.29	Tourism, some hunting	2012–2015	Zambezian Mopane Woodlands and Dry Miombo Woodlands	0.19	14	
Overall totals		2,864.17		2010–2016	Mean	0.44(0.06 S.E.)		
					Area Weighted Mean	0.48		
Notes.

1, Brassine & Parker (2015); 2, Boast et al. (2011); 3, Cheetah population size estimates in Kgalagadi and surrounding areas of south-western Botswana 2011–2015: Report to the Government of Botswana; D Keeping, 2016, unpublished data; 4, Maude (2014) extended analysis; 5, Maude (2014) extended analysis; 6, Maude (2014); 7, Broekhuis (2012); 8, Funston, Hanssen & Moeller (2014); 9, Institute for Zoo and Wildlife Research farmland survey 2012–2016; 10, Institute for Zoo and Wildlife Research farmland survey 2016; 11, Mills & Mills (2017); 12–14, Van der Meer (2016).

We applied these values to areas without observational data to identify potential cheetah range. Pixels below threshold values remained potential range whereas those above the threshold were removed. We then filtered three ecoregions within the historical range that are unlikely to contain resident individuals: Namib Desert, Kaokoveld Desert, and Makgadikgadi Halophytics. Although we did observe cheetahs in these ecoregions, they mostly occurred along the periphery of these areas and, historically, have been characterised as thinly scattered or only seasonally resident due to prey scarcity in these ecoregions (Myers, 1975; Klein, 2007). In the final step, we removed patches of potential habitat with less than 300 km2 (3 pixels) of core habitat where these patches are adjacent to areas excluded as cheetah habitat. We did so as our population analysis revealed that the weighted mean density of cheetah in the study area was 0.48/100 km2, determined using the empirical estimates described below, meaning that 300 km2 would support approximately one resident individual.

Density estimates

We searched the scientific literature for data recorded during the survey period that allowed estimates of cheetah densities. We collated published information with on-going surveys and re-analysed the data already published to increase sample sizes and improve accuracy. We excluded repeat studies of the same areas, and considered only the most recent results. This resulted in 14 empirical estimates of cheetah density (Table 1).

To estimate the total regional population of cheetahs, we stratified cheetah presence pixels (including the buffer) by ecoregion. In each ecoregion, we assigned a density value based on the weighted mean of empirical estimates for the ecoregion shown in Table 2. The estimate for the Namib Desert was applied to all other deserts and halophytic ecoregions. We used an average of the estimates from the Namib Desert and the Kalahari Xeric Savannah for the Gariep Karoo which lies geographically between these two. For the Zambezian Flooded Grasslands we applied the density estimate from the adjacent Zambezian Baikiaea Woodlands. In all other ecoregions without empirical estimates, we applied the weighted mean of all empirical density estimates (0.48/100 km2).

Table 2 Numbers and densities of free-range cheetahs.

Location ecoregions	Presence area* (100 km2)	Possible presence area (100 km2)	Inferred density	Cheetah population	Possible additional cheetah population	Footnote	
Direct estimates							
Zimbabwe	825			160		a	
Kruger NP	168			412		b	
Indirect estimates							
Kalahari Xeric Savanna	2,738	3,166	0.53	1,451	1,615	c	
Angolan Mopane Woodlands	996	385	0.48	478	181	d	
Kalahari Acacia Woodlands	616	444	0.48	296	209	d	
Namibian Savannah Woodlands	480	95	0.20	96	19	e	
Namib Desert	396		0.20	79	0	e	
Gariep Karoo	333	1,575	0.36	120	567	f	
Central Bushveld	317	59	0.48	152	28	d	
Zambezian mopane woodlands	265	531	0.51	135	271	g	
Zambezian Baikiaea Woodlands	251	776	0.18	45	140	h	
Kaokoveld Desert	153		0.20	31	0	e	
Zambezian Flooded Grasslands	112	137	0.18	20	25	h	
Limpopo Lowveld	79		0.48	38	0	d	
Etosha Pan Halophytics	48		0.20	10	0	e	
Albany Thickets	29		0.48	14	0	d	
Namaqualand-Richtersveld Steppe	29	235	0.48	14	110	d	
Highveld Grasslands	17		0.48	8	0	d	
Nama Karoo Shrublands	14	13	0.48	7	6	d	
Makgadikgadi Halophytics	13		0.20	3	0	e	
Miscellaneous habitats (<10,000 km2)	18	12	0.48	9	6	d	
Totals	7,897	7,428		3,577	3,250		
Notes.

a From Van der Meer (2016), who found cheetahs mostly in areas of Zambezian Baikiaea and Mopane Woodlands ecoregions (see Fig. 1).

b From Marnewick et al. (2014). Kruger NP is classified as mostly Mopane Woodlands.

c Density is a weighted average of estimate #s 2, 3, 4, 5, 6, 9 and 11 from Table 1.

d We have no specific estimates of cheetah densities for this ecoregion, however we know this is a highly suitable habitat, so we use the overall weighted density estimate from Table 1.

e We used the density estimate # 10 from Table 1.

f We used the average density of Kalahari Xeric Savanna and Namib Desert under the assumption that this ecoregion should have an intermediate density.

g Density is a weighted average of estimate #s 1, 7, 13 from Table 1. Density sample 14 also contains Zambezian Mopane Woodlands but this sample seems to be more representative of the Dry Miombo ecoregion in Zimbabwe, already accounted for in Van der Meer (2016).

h Density is a weighted average of estimate #s 8 and 12 from Table 1.

* Areas include buffers (see text).

Figure 1 Cheetah distribution in the study area in southern Africa.

We used existing cheetah population estimates for Zimbabwe (Van der Meer, 2016) and Kruger National Park (Marnewick et al., 2014). We calculated per pixel (100 km2) density estimates for both areas to compare cheetah population density to the rest of the study area. In Kruger, we determined per pixel density using the estimated cheetah count and park area, assuming consistent population density. However, we could not assume that cheetah density in Zimbabwe was consistent across known cheetah presence pixels. Therefore, we calculated an estimate of cheetah density for each ecoregion with cheetah presence in Zimbabwe using available count data from Van der Meer (2016).

If Di is the estimated density of cheetahs for ecoregion i outside Zimbabwe, then the density of cheetahs in ecoregion i inside Zimbabwe (Di′) is calculated by multiplying Di by the ratio of the Van der Meer (2016) cheetah count (Pz) to the sum of ecoregion population estimates, calculated as density (Dj) times area (Aj) of n ecoregions with cheetah presence in Zimbabwe. Di′=DiPz ∑j=1nDjAj.

For pixels in possible cheetah range, we assigned cheetah densities using the same ecoregion approach we used in confirmed cheetah presence areas, detailed in the section above.

Data sources for off-take estimates

We defined persecution as the effective removal—off-take—of cheetahs from the free-ranging population via lethal control or permanent captivity. During the assessment period, we recorded details of cheetah persecution on 185 commercial farmland properties across nine regions in Namibia over an area of 19,184 km2 (median size = 65.5 km2), or approximately 5.4% of the commercial farmland of the country (Mendelsohn, 2006). Persecution data were recorded during direct, on-site carnivore consultations with land managers as part of a conflict research programme. The land use and management characteristics recorded for this sample were similar to those previously reported for commercial farmland across Namibia (Mendelsohn, 2006; Lindsey et al., 2013a; Lindsey et al., 2013b) (Appendix S5). Persecution data usually included information on age and sex of the cheetah (Appendix S6).

Leslie Matrix model

Leslie Matrix models calculate growth rates for age-structured populations and so require information on several life history parameters (Caswell, 2001). These models have varied practical applications, including assessing management options for highly threatened species (Fujiwara & Caswell, 2001). We used these models to estimate by how many females the population can be reduced per year while still permitting a constant population size over time. We then compared these results with persecution data.

We employed a simple model that required only the age at first reproduction, inter-birth interval, number of offspring that reached adulthood, and adult survival rates. We searched the literature for all relevant life history data. We review the parameters gleaned from the literature below.

In Serengeti National Park, Tanzania, Kelly et al. (1998) estimated the age of first reproduction at 2.4 years (29 months), essentially two years plus the estimated 90 to 95 day gestation period known from both captive and free-ranging cheetahs (Brown et al., 1996; Eaton, 1974). Kelly et al. (1998) estimated the inter-birth interval at 20.1 months (n = 36) whereas Marker et al. (2003) reported a range of 21–28 months (mean = 24, n = 6) for Namibian farmland.

The number of offspring reaching independence (at approximately 17 months) varied more substantially across data sources (Laurenson, 1992; Laurenson, 1994; Laurenson, Wielebnowski & Caro, 1995; Kelly et al., 1998). Some studies observed juveniles from their detection in the lair to independence, whereas other studies observed offspring detected at any age to independence (Frame & Frame, 1976; McVittie, 1979; Morsbach, 1986a; Morsbach, 1986b; Marker et al., 2003; Pettorelli & Durant, 2007; Marnewick et al., 2009; Wachter et al., 2011; Mills & Mills, 2014; Weise et al., 2015). The presence of carnivore species, particularly large ones such as lions Panthera leo and spotted hyenas Crocuta crocuta, can be a major factor affecting offspring survival (Laurenson, 1994; Wachter et al., 2011; Mills & Mills, 2014). For Namibian farmland without these species, the range of young cheetahs raised to independence varied from 1.3 offspring per litter (Marker et al., 2003) to 3.2 offspring per litter of an average litter size of 4.7 surviving to 14 months (Wachter et al., 2011). In the Kgalagadi Transfrontier Park, Mills & Mills (2014) estimated 1.5 offspring per litter surviving to independence (45% of an average litter size of 3.4).

Adult female survival was also reported in different ways either as averaged life spans or as annual survivorship. Kelly et al. (1998) recorded an average life span of 6.2 years for the Serengeti National Park, or an estimated annual survivorship of 89.4% whereas Marnewick et al. (2009) estimated an annual survivorship of 88.6%, corresponding with an average lifetime of 5.7 years for the Kruger National Park.

We ran the Leslie Matrix model using various combinations of life history parameters, based on the literature outlined above, to test their sensitivity to changing predicted annual growth rates. We created an optimized model with the parameters that would result in the highest growth rates, and then subsequent models resulting in lower growth rates were used for life history parameters using the variation reported in the literature.

These models consider only female population growth rates. The model assumed there will always be sufficient males to breed with all females, thus we did not separately model males. The model is implemented in a Microsoft Excel spreadsheet (Appendix S7).

Once the growth rates for the female cheetah population were determined using the Leslie Matrix models, we calculated the stable cheetah population density Ds (the density at which known off-take does not result in population decrease). Ds=2OfAsλf.

We determine Ds using λf, the female cheetah population growth rate as determined by the Leslie Matrix model, Of, the off-take number of female cheetahs, and As the study area across which off-take is determined, in this case 19,184 km2. Following the sex ratio of young adult cheetahs at independence in the Kalahari (Mills & Mills, 2017), we assume a 1:1 sex ratio of females to males in the final population.

Results

Cheetah presence observations

Most cheetah presences came from research data (Table 3; Fig. S1). Crowd-sourced point data uniquely contributed 12.9% of presence pixels of free-ranging cheetahs and corroborated an additional 10.8% of presence pixels. 69.2% of pixels attributed to crowd-sourced data were in IUCN categories I–IV protected areas and an additional 13.7% were in other protected areas. In contrast, research data were found primarily outside protected areas with only 18.9% found in IUCN I–IV protected areas and an additional 10.7% in other protected areas.

Table 3 Area (in 100 km2) of data contributions per country and within protected areas.

	Research data	Research data in protected areas IUCN I–IV (% of total research)	Research data in all protected areas (% of total research)	Crowd- sourced data	Crowd-sourced data in protected areas IUCN I–IV (% of total crowd sourced)	Crowd-sourced data in all protected areas (% of total crowd sourced)	Corroborated data (i.e., both sources)	Corroborated data in protected areas IUCN I–IV (% of total corroborated)	Corroborated data in all protected areas (% of total corroborated)	
Botswana	388	105 (27.1%)	105 (27.1%)	27	19 (70.4%)	19 (70.4%)	26	18 (69.2%)	18 (69.2%)	
Namibia	767	117 (15.3%)	190 (24.8%)	34	23 (67.6%)	28 (82.4%)	8	4 (50.0%)	5 (62.5%)	
South Africa	117	11 (9.4%)	42 (35.9%)	140	105 (75.0%)	126 (90.0%)	40	31 (77.5%)	39 (97.5%)	
Zimbabwe	148	36 (24.3%)	84 (56.8%)	40	20 (50%)	27 (67.5%)	127	45 (35.4%)	78 (61.4%)	
Totals	1,420 (76.2%)			241 (12.9%)			201 (10.8%)			

Table 4 Area of cheetah distribution (in 100 km2) across countries and protected areas.

	Botswana	Namibia	South Africa	Zimbabwe	Total study aea	Protected areas IUCN I–IV	All Protected Areas (I–VII)	Kavango- Zambezi (KAZA) a	
Free range cheetah presence	441 (11.1%)	2,897 (73.2%)	289 (7.3%)	333 (8.4%)	3,960	605 (15.3%)	2,353 (59.4%)	562 (14.2%)	
Presence buffer	1,297 (32.9%)	1,497 (38.0%)	652 (16.6%)	492 (12.5%)	3,938	870 (22.1%)	1,297 (32.9%)	515 (13.1%)	
Managed metapopulation	0	0	130 (100.0%)	0	130	6 (4.6%)	46 (35.4%)	0	
Possible cheetah presence	3,069 (41.3%)	2,956 (39.8%)	1,403 (18.9%)	NA	7,428	738 (9.9%)	1,066 (14.4%)	1,284 (17.3%)	
Total cheetah presence area without metapopulation	1,738 (22.0%)	4,394 (55.6%)	941 (11.9%)	825 (10.4%)	7,898	1,475 (18.7%)	3,650 (46.2%)	1,077 (13.6%)	
Total presence area with metapopulation	1,738 (21.6%)	4,394 (54.7%)	1,071 (13.3%)	825 (10.3%)	8,028	1,481 (18.4%)	3,696 (46.0%)	1,077 (13.4%)	
Total cheetah presence area with possible presence areas	4,807 (31.1%)	7,350 (47.6%)	2,474 (16.0%)	825 (5.3%)	15,456	2,219 (14.4%)	4,762 (30.8%)	2,361 (15.3%)	
Percent area with cheetah presence (including managed metapopulation)	30.0%	53.5%	7.7% (8.8%)	21.1%	26.2% (26.7%)				
Notes.

a Figure S2 shows the Kavango-Zambezi transfrontier conservation area (KaZa TFCA) overlaid on cheetah distribution.

Range

Cheetah presence in free-range habitat encompassed 789,800 km2 of the study region (Table 4; Fig. 1), including the buffer around verified presence. The largest proportion of the total verified cheetah range occurred in Namibia (55.6%), the least in Zimbabwe (10.4%) (Table 4). The country with the greatest proportion of total surface area occupied by cheetah range was Namibia (53.5%). Of the current known free-range in southern Africa, 18.4% is formally protected (IUCN categories I–IV) and an additional 27.6% by the remaining categories (V–VII). Occurrence records suggest that cheetah populations in these four countries are linked across international boundaries (Fig. 1) and 13.6% of the documented free-range presence falls into the Kavango Zambezi Transfrontier Conservation Area.

In South Africa, small fenced reserves across nine provinces comprised a managed meta-population, with a total size of 11,721 km2 (Fig. 1). Note, due to our spatial resolution, the estimated area is slightly larger than this at approximately 13,000 km2.

Cheetahs are generally not observed in areas with high human and livestock densities thus we applied thresholds across the remainder of the study area where no presence information was available to determine possible presence areas. Possible cheetah presence areas comprised another 742,800 km2 of the study area (Fig. 2). In contrast, most of South Africa, eastern Botswana, and the northern part of Namibia adjacent to Angola are above these thresholds, suggesting cheetahs would be absent.

We confirmed free-ranging cheetahs across 789,800 km2 based on verifiable observations; when the fenced population is included this increased to 802,800 km2. Including possible presence areas increased the cheetah range to 1,545,600 km2.

Densities

We sourced 14 empirically determined local to regional cheetah density estimates, covering 286,417 km2, or approximately 36% of the area known to support free-ranging cheetahs (Fig. 3A; Table 1). Estimated densities varied from 0.18–0.90 individuals per 100 km2. The mean density across study sites was 0.44 ± 0.06 S.E. cheetahs per 100 km2. Weighting the mean by the area surveyed (mean calculated from total number of 100 km2 sample blocks with measured density) yielded an overall density of 0.48 cheetahs per 100 km2.

Under managed conditions, the densities on small fenced reserves in South Africa ranged from 0.11–15.0 individuals per 100 km2 (Appendix S8). Nearly all reserves contained at least 1.0 cheetah per 100 km2 (88.0%, n = 44) while 21 reserves contained 5.0 cheetah per 100 km2 or more. Only 14 of the reserves reported juveniles, hence evidence for breeding. The densities from managed reserves are not included in the calculations of free-ranging populations.

Including densities based on cheetah counts in Zimbabwe and Kruger National Park, the density of free-ranging cheetahs varies from 0.09 per 100 km2 in the Dry Miombo and Zambezian Baikiaea Woodlands to more than 2.0 per 100 km2 in the Kruger National Park (Fig. 3B). When study density estimates were applied across ecoregions in presence areas without cheetah monitoring, the minimum estimated density was 0.18 per 100 km2 in Baikiaea Woodlands and Flooded Grasslands and the maximum estimated density was 0.51 per 100 km2 in Kalahari Xeric Savanna and Zambezian Mopane Woodlands, with a weighted mean density of 0.36 cheetahs per 100 km2 in the same area. By comparison, the IUCN status assessment implies an average density of approximately 0.35 adults per 100 km2.

Figure 2 Known cheetah presence in relation to human and livestock densities.

(A) Human population per sq. km. (B) Cattle per sq. km. (C) Goats per sq. km. (D) Sheep per sq. km.

Figure 3 Cheetah densities within the study area.

(A) Locations of cheetah density estimates overlaid on the major ecosystem types in the study area. (B) Estimated cheetah densities in presence and possible presence areas. See Table 3 for the source of the density estimates.

Population

Based on known (Table 1) and inferred densities calibrated to ecoregion types (Table 2), we estimated 3,577 free-ranging adult cheetahs in southern Africa with a maximum additional 3,250 cheetahs in potential habitat areas. At the end of July 2016, 176 adult cheetahs lived in small fenced reserves in South Africa.

Our estimates of free-ranging cheetah numbers are of three kinds. First, across Zimbabwe Van der Meer (2016) estimated 150-170 adults, of which 104 were individually recognized as 52 males, 30 females, 22 of unknown sex, plus approximately 60 offspring. Using this study, we estimate 160 resident individuals in Zimbabwe. Marnewick et al. (2014) estimated 412 adults in Kruger National Park in South Africa from 2008 to 2009. This falls outside our study period but the count was included because we consider it the most reliable estimate of cheetahs in this area. Second, for all other confirmed cheetah presence pixels we applied cheetah densities based on ecoregion (Table 2; Fig. 3B). We predict approximately 3,005 cheetahs in these areas. We estimated the highest number of cheetahs (1,451 individuals) in the Kalahari Xeric Savannah ecoregion which covers 273,800 km2 of connected habitat in Namibia, Botswana, and north-western South Africa (Fig. 3A). The second highest number was 478 animals in the Angolan Mopane Woodlands covering 99,600 km2. Third, and not included in the estimate of 3,577 individuals, another 742,800 km2 may hold cheetahs. This possible range spans ecoregions with densities ranging from 0.18 to 0.51 cheetahs per 100 km2 (Fig. 3B). If cheetah fully occupied possible range at the same densities as known cheetah presence areas, this would add another approximately 3,250 animals, suggesting a maximum adult population of 6,827 individuals in the four study countries.

Persecution

On Namibian farms (n = 185), 26.5% of land managers actively persecuted cheetahs while 49.7% considered the species as causing conflict (Appendix S6). On these properties, managers trapped a total of 245 cheetahs during the survey period, of which 17 were translocated (Weise et al., 2015), 32 were placed into permanent captivity, and 196 were killed (146 verified plus 50 reported). This resulted in an effective annual removal of 0.59 cheetahs per 100 km2 over all ages and sexes, 0.30 adult cheetahs per 100 km2 per year, including 0.10 breeding age females (Appendix S6). Persecution was skewed towards adult males (32.9% of all 146 aged and sexed animals) and sub-adult males (26.7%) compared to adult females (17.1%) and sub-adult females (23.3%), but not significantly different across ages and sexes (χ2 = 3.51, d.f. = 3, p = 0.318).

The primary income sources of the farm managers influenced levels of cheetah persecution. Using documented persecution levels as a proxy for tolerance for the species, commercial wildlife farming and hunting operators had a disproportionately high impact on cheetah removal, while recreational land uses were the most tolerant (χ2 = 41.2, d.f. = 4, p < 0.001). The few least-tolerant land managers had a disproportionately high impact on cheetah removal. Ten farm owners removed 71.9% of all persecuted cheetahs; possibly inducing local population sinks. The three most intolerant managers (two wildlife ranchers and one cattle owner) contributed 50.0% to persecution, including one manager accounting for 36.0% of all removals (Appendix S6).

Estimating densities from persecution data

Leslie Matrix models predict population growth rates under various assumptions about key demographic parameters. The models can uncover which of those parameters changes the growth rates the most and so sometimes provide key insights into the species’ management. For an exploited population, the models predict how many individuals can be removed without causing the population to decline. Conversely, if one knows the numbers of animals removed from a population and its growth rate, one can estimate the species’ population size. Notice, that the greatest possible growth rate corresponds to the lowest densities of cheetahs that can support a given level of persecution without causing the population to decline. We estimate growth rates and calculate minimum densities of cheetah needed to support known persecution levels.

First, we use the most optimistic scenario of demographic factors influencing population growth in the Leslie Matrix model: 29 months at first reproduction, a 20.1 months inter-birth interval, 3.2 offspring per litter raised to independence, and a 6.2 year life span for adult females. With these parameters, cheetah populations can grow at 29.9% per year (Table 5). Based on observed cheetah persecution rates in Namibia and, assuming the most optimistic scenario of growth rates, a density of 0.67 reproductive cheetahs per 100 km2 would sustain the known persecution rate of 0.1 females per 100 km2 (approximately 19.2 female cheetahs in the Namibian study area) per year without population decline.

Table 5 Leslie Matrix parameters and model outputs.

Input parameters	Model 1	Model 2	Model 3	Model 4	
Age at first reproduction (months)	29a	29	29	29	
Litter size	5b	5	5	5	
Interbirth interval (months)	20a	24e	24	24	
Adult survival annual (%)	89.4a	89.4	88.6f	88.6	
Average life span as adults (years)	6.2a	6.2	5.7	5.7	
Age at independence (months)	17c	17	17	17	
Number of cubs raised to independence	3.2d	3.2	3.2	2.3g	
Model outputs					
Growth Rate Estimate (%)	29.9	25.2	24.6	12.4	
Inferred Stable Cheetah Population Density (per 100 km2)*	0.67	0.79	0.81	1.61	
Notes.

a Kelly et al. (1998).

b Intializing model assumption based on estimates of litter size in Namibia, 4.7 ± 0.9, from Wachter et al. (2011).

c Laurenson (1992), Laurenson (1994), Laurenson, Wielebnowski & Caro (1995) and Kelly et al. (1998).

d Wachter et al. (2011).

e Marker et al. (2003).

f Marnewick et al. (2009).

g Mills & Mills (2014).

* Assuming 1:1 sex ratio and .1/100 km2 female cheetah offtake rate.

Second, for models with lower growth rates, a higher density of animals would be required to sustain the population. When manipulating a single parameter at a time, the growth rate fell to 25.2% when the inter-birth interval increased to 24 months, to 24.6% when a reduced life span of 5.7 years was used, and to 12.4% when only 2.3 offspring per litter survive to independence. Were the growth rate 12.4%, 1.61 reproductive cheetahs per 100 km2 would sustain the known persecution rate (0.1 females per 100 km2).

Another way to view these results is that with the most optimistic parameters and a density of 0.67 adult cheetahs (assuming a one to one sex ratio) per 100 km2, cheetah density can be maintained given the known persecution. The density would be among the highest recorded in the region, suggesting that only under the very best conditions can cheetahs withstand persecution. Likely, elsewhere where conditions are not favourable, intensive persecution would eliminate them.

Discussion

Our objective was to provide independent estimates of cheetah distribution and abundance in southern Africa, considering additional data sources and processes not often used for this purpose, e.g., crowd-sourced information, estimates of cheetah persecution, and maps of human impact.

Our population estimate for cheetah range is lower than that produced by the IUCN/SSC (2015). In Zimbabwe, both studies relied on Van der Meer (2016) and we found few additional data using alternative sources. Our assessment of “confirmed” cheetah range relied only on verifiable cheetah observations, resulting in a smaller estimate of known cheetah distribution than that proposed by the IUCN assessment, highlighting the areas in which expert opinion form the basis for proposed cheetah range and for which we were unable to obtain observation data. Our population estimate of approximately 3,577 adult cheetahs is 11% less than the 4,029 adults estimated by the IUCN/SSC (2015), supporting Durant et al.’s (2017) call for up-listing the cheetah to “endangered” status. If we assume the same ecoregion-based cheetah densities in possible cheetah presence areas, our overall estimate would rise to approximately 6,827 adults. As it is very unlikely that all these possible presence areas contain cheetahs at the same density as confirmed cheetah presence areas, we urge greater caution in applying the upper end of our population estimate as opposed to its low limit, based on only those areas with confirmed observations.

While the differences between our estimate of cheetah distribution and that produced by the IUCN/SSC (2015) may appear small, they have important implications for conservation. We estimated the known cheetah range to be 789,800 km2 in the four countries (802,800 km2 when including managed reserves)—an area based exclusively on confirmed data but which included an adjacency buffer around verified free-range presence. We speculate that cheetahs may occur across another 742,800 km2 due to suitable habitat and low human and livestock densities, resulting in a total possible range of 1,545,600 km2. For the same four countries, Durant et al. (2017) estimated 1,149,000 km2 of confirmed and 245,000 km2 of possible presence, a total of 1,394,000 km2. Much of this difference arises from Durant et al. (2017) using expert opinion to inform areas where data are sparse (Fig. S3). While this is understandable, particularly for protected areas, using an expert system approach to range mapping raises issues about supporting evidence. On the other hand, we appreciate that our approach is correlative and does not provide causal evidence to indicate why cheetahs may or may not live at certain densities. In addition, the global data set of livestock densities (Robinson et al., 2014) may have inaccuracies at local scales.

An important difference between our study and the RWCP process is in how we choose to present the data, which include many sensitive records. In addition to summarised GPS records from collared individuals, we compiled nearly 20,000 observations, and aggregated them at a 10 x 10 km grid. Reducing the resolution of the observations allowed us to publicly display all the input data (Fig. 1). IUCN maps do not provide this level of detail to the public, although this information is collected by the RWCP and it is available upon request. The difference in presentation has some important consequences in framing questions for research that may drive future assessments of the species’ status.

First, our approach is explicit about the sampling bias. This allows us to understand where estimates are derived from research and where estimates are based on expert opinion. Across much of central and eastern Botswana there are only scattered observations (Fig. 1). Given how sparsely populated and inaccessible much of central Botswana is, it is perhaps sensible to presume that the species is present throughout this area. Nonetheless, not explicitly linking presence data to potential range may have the effect of discouraging surveys in places where presence is only assumed. The IUCN map also occasionally extends the cheetah’s range 100–200 km outside our known records. It is possible that we may have missed data supporting these extensions, but if not, verified observations and new surveys in these areas would be most important. We propose that the commercial farmland in south-eastern Namibia and northern South Africa, and the farmland in north-western, central, and eastern Botswana are areas of particular research interest in determining distribution.

Second, our approach permits discussion about where cheetahs might be and we can ask detailed questions concerning the uncertainty of our analyses. For example, adding all possible areas of cheetah presence more than doubles our population estimate. This is an unlikely scenario; hence, this upper estimate serves to highlight the need for further research in such areas rather than providing a realistic assessment of the species’ status. Another important uncertainty stems from the few observations in central Botswana in the Kalahari Xeric Savannah and Kalahari Acacia Woodland ecoregions. Documenting presence here is important; based on the ecoregion densities this area could contain a maximum of approximately 1,100 cheetahs. Other important areas are southern Namibia and northern South Africa, where habitat in Gariep Karoo and Kalahari Xeric Savannah could support another 1,200 cheetahs.

A final set of uncertainties arise from the study being large-scale, multi-year, and retrospective. We have no control over the individual survey designs and so cannot address statistical assumptions including detection probabilities and whether the population as we have defined it is closed to immigration or emigration. Such concerns apply to all large-scale range wide surveys, of course.

Third, as a corollary to evaluating assumptions about where cheetahs might be, one may evaluate our assumptions about where they are unlikely to be found. While the absence of evidence for cheetahs should not be automatically equated with evidence for their absence, combined with other data (e.g., Fig. 2) it can be suggestive. High densities of humans and their livestock likely preclude permanent cheetah presence and we excluded these areas from possible range. Reliable observations in such areas would be important in confirming or refuting this assumption, as well as for building a better understanding of the factors that restrict the range of cheetahs. Such exercises are beyond the scope of this paper, but further refinements are necessary at national scales. Similarly, there is a need to calibrate local density estimates to the cheetah’s complex spatial ecology (J Melzheimer, 2002–2014, unpublished data).

Fourth, we found that known cheetah populations are remarkably concentrated. About 55% of the known population are located within approximately 400,000 km2, consisting of 259,600 km2 of the Kalahari Xeric Savannah and 143,600 km2 of the Namibian Savannah Woodlands. Thus, while cheetah range may be contiguous across the four study countries in southern Africa, most individuals live in a particular portion of that range. Much of the Kalahari Xeric Savannah overlaps with privately held farmland, an area of higher risk for human-wildlife conflict and associated persecution. Within the core high density cheetah range in Kalahari Xeric Savannah between Kgalagadi Transfrontier Park and Central Kalahari Game Reserve (see Fig. S2, Fig. 3B), Botswana plans to relinquish approximately 8,230 km2 of Wildlife Management Areas for expanded livestock production (Government of Botswana, 2009). The continued large-scale conversion of conservation lands will almost certainly exacerbate conflict and negatively impact the southern African free-ranging cheetah population.

Value of crowd-sourced data

We provide an extensive and replicable process to gather data from public sources (Appendix S1). Reliable crowd-sourced information uniquely contributed 12.9% to our distribution estimate, sometimes being the only available information for specific areas (e.g., Etosha National Park in Namibia). Furthermore, it was essential in assessing cheetah status in Zimbabwe (Van der Meer, 2016). Nevertheless, we should not expect crowd-sourced data across all of cheetah range. Such data did add to our knowledge, but originated largely from protected areas and within areas the RWCP process classified as extant range (85.3% of crowd-sourced observations were within extant range, and 76.9% from protected areas). Indeed, the crowd-sourced data were even more restricted, being primarily available for protected areas with high visitor volumes. Knowing the patterns of crowd-sourced data can be beneficial in understanding biases in our assessment processes (Boakes et al., 2016). For the cheetah, obtaining crowd-sourced data can assist in assessing numbers in parks with high tourist volumes (see Marnewick et al., 2014), and could free up valuable research effort to focus on unprotected lands where most cheetah occur and are more vulnerable to anthropogenic threats. We encourage the use of citizen science particularly for protected areas.

Off-take data and the Leslie Matrix model

We provide some of the most detailed off-take records for large carnivores. These data are from commercial farmland in Namibia and are not applicable to all areas of range in southern Africa (e.g., protected areas). Nevertheless, our data indicate the importance of documenting persecution hotspots because even a single land manager can eliminate large numbers of animals (more than one-third of all documented persecution). High persecution levels recorded on Namibia’s farmlands corresponded with the highest cheetah density (0.7/100 km2) recorded outside protected areas (Fig. 3A, Table 1). In southern Africa, livestock and wildlife farmlands provide large swaths of habitat for the cheetah outside protected area (e.g., Thorn et al., 2010; Boast & Houser, 2012; Lindsey et al., 2013a; Williams et al., 2016). The ten farmland properties that accounted for nearly 72% of the off-take records had commercially stocked wildlife or livestock. Here, managers attach high monetary value to wild and domestic cheetah prey (e.g., Lindsey et al., 2013b; Weise, 2016). Even a few stock losses may exceed local tolerance for conflict and trigger intensive persecution (e.g., Thorn et al., 2010; Lindsey et al., 2013a; Weise, 2016). Indeed, approximately 15% of our confirmed cheetah range overlaps with human and livestock densities higher than the thresholds we used to infer potential cheetah range, implying high potential for conflict with human interests in these areas. Focusing future conservation efforts on known persecution hotspots, and those areas with highest potential for conflict, may help prevent continued unsustainable removal, as otherwise, locally concentrated persecution may continue to inflict substantial losses.

Secondly, these detailed data allow us to estimate cheetah population densities required to sustain these levels of off-take without population decline. For the Kalahari Xeric Savannah, the Leslie Matrix model suggested a density of 0.67 adult cheetahs per 100 km2 (at equal sex ratio) was necessary to support the level of persecution we observed. This was only slightly lower than the density recorded in this area (0.7; Table 1).

The density estimate based on persecution would have been substantially higher had we used demographic parameters that were typical for cheetahs rather than their optimal ones for growth (e.g., as high as 1.61). Were that the case, one explanation might be that there are far more cheetahs in this area than currently recognised. This seems unlikely as there are respected density estimates from this region that are only ∼0.7. Another alternative may be that the high off-take is causing declining cheetah densities. Our persecution data may involve animals drawn into local sink habitat on Namibian farms, i.e., through the continued removal of resident cheetahs from high quality environment with reduced intra-guild competition and abundant prey (e.g., see Lindsey et al., 2013a; Lindsey et al., 2013b), meaning an area larger than the sampled properties is supporting this loss. Individual properties with high off-take levels may act as attractive sinks that potentially induce source–sink dynamics (Battin, 2004). In addition, the high numbers of animals killed in Namibia possibly reflect a period when cheetahs might have been unusually abundant because of above average rainfalls (Climate Change Knowledge Portal, 2017), supporting high prey densities between 2009 and 2012 (e.g., Lindsey et al., 2013b). These considerations underline the uncertainties of assuming an overall and constant density estimate. The most parsimonious explanation is that the close similarity of the two estimates (0.67, 0.7) suggests that cheetahs are, at best, holding their own in an area of relatively high productivity in the face of intense persecution.

Conclusion

Our independent assessment, with an estimated range size lower than the IUCN estimate and an estimated adult cheetah population of approximately 3,577 free-ranging animals, supports the conclusion of Durant et al. (2017) to review the cheetah’s threat status and consider up-listing the species to endangered status.

Our results also demonstrate how concentrated cheetah records are within southern Africa. A mere 400,000 km2 contain approximately 55% of the known population, much of it on unprotected lands. This area corresponds with high levels of persecution, thus generating a concern that the stronghold is at risk of ‘hollowing out’ and highlights how precarious the situation is for this species. We also show the impact of a few landowners on overall cheetah removal, suggesting that focused conservation efforts in known persecution hotspots could substantially reduce off-take.

Supplemental Information

Figure S1 Distribution of research and crowd sourced presence observations in the study area in southern Africa

Click here for additional data file.

Figure S2 Overlap of cheetah distribution with the Kavango Zambezi Transfrontier Conservation Area (KAZA)

Click here for additional data file.

Figure S3 Comparison of the cheetah distribution detailed here with that published by the IUCN

Click here for additional data file.

Appendix 1 Public data search methodology

Click here for additional data file.

Appendix 2 Known cheetah lifespans in southern Africa

Click here for additional data file.

Appendix 3 Additional detail on presence data and distribution mapping

Click here for additional data file.

Appendix 4 Relationship of cheetah presence locations with covariates i.e., human and livestock densities

Click here for additional data file.

Appendix 5 Farm characteristics where cheetah persecution occurred

Click here for additional data file.

Appendix 6 Details of cheetah persecution data in Namibia

Click here for additional data file.

Appendix 7 Implementation of the Leslie Matrix model

Click here for additional data file.

Appendix 8 Cheetah metapopulation data in South Africa

Click here for additional data file.

For their assistance, suggestions, and data we sincerely thank J Wilson, C Winterbach, N Mitchell, A Brassine, D Parker, A Stein, S Durant, R Portas, V Menges, I Wiesel, K Stratford, H Davies-Mostert, L Hanssen, H Van der Meer, AM Houser, T Wassenaar, NAPHA Namibia, K-U Denker, T Dahl, D Van der Westhuyzen, B Wasiolka, J Power, D Ward, C R Kotze, MET Namibia, K Uiseb, P Beytell, O Aschenborn, DWNP Botswana, M Flyman, SANBI, C Brain, L Boast, L Van der Weyde, D Cilliers and Cheetah Outreach Trust SA, D Willuhn, C Meier-Zwicky, K-H Wollert, M Graben, K Stäber, N Odendaal, Greater Sossus-Namib Landscape Initiative, K Echement, R Thompson, M Keith, as well as all hunters and private land owners who contributed to this survey. We thank the CCB research team and Duke University students A Bennett, K Malloy, E Mills, R Hinson, H Punjani, P Ranganathan, D Tarrazo, K Vayda, and A Wong. We also thank the Research Council of Zimbabwe and Zimbabwe Parks and Wildlife Management Authority for permits and support. Similarly, we thank the Botswana Ministry of Environment, Wildlife and Tourism and Namibia’s Ministry of Environment and Tourism for granting permission to undertake research in these countries.

Additional Information and Declarations

Competing Interests

Author Contributions

Data Availability

Stuart Pimm is an Academic Editor for PeerJ.

Florian J. Weise conceived and designed the experiments, performed the experiments, analyzed the data, contributed reagents/materials/analysis tools, wrote the paper, reviewed drafts of the paper.

Varsha Vijay and Stuart L. Pimm conceived and designed the experiments, performed the experiments, analyzed the data, contributed reagents/materials/analysis tools, wrote the paper, prepared figures and/or tables, reviewed drafts of the paper.

Andrew P. Jacobson performed the experiments, contributed reagents/materials/analysis tools, wrote the paper, prepared figures and/or tables, reviewed drafts of the paper.

Rebecca F. Schoonover performed the experiments, analyzed the data, contributed reagents/materials/analysis tools, wrote the paper, prepared figures and/or tables, reviewed drafts of the paper.

Rosemary J. Groom, Jane Horgan, Derek Keeping, Rebecca Klein, Kelly Marnewick, Glyn Maude, Jörg Melzheimer, Gus Mills, Vincent van der Merwe, Esther van der Meer Rudie J. van Vuuren and Bettina Wachter performed the experiments, wrote the paper, reviewed drafts of the paper.

The following information was supplied regarding data availability:

Dryad; doi: 10.5061/dryad.7n4h4.

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
