# Peer review of "The distribution and numbers of cheetah (Acinonyx jubatus) in southern Africa"

_PeerJ, doi:10.7717/peerj.4096_

## Round 0.1 · original submission · Major Revisions

The authors did an impressive work with a large amount of data and a standard methodology for assessing the range and numbers of cheetah.

Before I approve it for publication, I would like the authors to improve the manuscript based on the reviewers’ comments.

I look forward to receiving your revised manuscript.

Best regards,

Hugo

Reviewer 1 ·

Basic reporting

I am in agreement with the authors that presence data should be made more public as this makes it easier to understand and interpret species distribution. At the same time it is also important to provide enough information so that the study can be replicated and I feel that this needs more attention by the authors. For example, the article could do with some restructuring as some of the methods are presented in the results section. It would also helpful if similar ideas are grouped together and put in the same orders (i.e. results presented in the same order as the methods). At the moment some section jump around a bit making it difficult to follow the authors' train of thought.

Experimental design

The authors have collected an incredible amount of data and it is clear that they have put a lot of time and effort into the study. However, there are some concerns regarding the extrapolation of density figures, especially as most figures used violate various assumptions incl. closure and do not account for detection probability. Also, the methods could do with more detail so that the study can be replicated (details are provided in the comments).

Validity of the findings

One of the major concerns is that no measure of precision is provided for the overall number of cheetahs and therefore more care should be taken when interpreting results and comparing these to other findings.

Annotated reviews are not available for download in order to protect the identity of reviewers who chose to remain anonymous.

·

Basic reporting

All my comments are included in the "general comments".

Experimental design

All my comments are included in the "general comments".

Validity of the findings

All my comments are included in the "general comments".

Additional comments

I applaud the authors for this impressive work, including large amount of data and the attempt of introducing a standard methodology for assessing the range and numbers of cheetah in South Africa. As compared to previously available data on the distribution of the species, this paper provides an updated status at high resolution as well as the regional variation in cheetah densities, which is based on a rich and diverse sources of data. It also attempts to determine the possible range of the cheetah based on an assumption of negative relationship between the presence of a large predator and anthropogenic factors. I think, as such, the results presented in this manuscript are generally a valuable contribution to the science as it provides a detailed information on the range and numbers of a rare (difficult to survey) species supported with abundant and statistically elaborated data.
However, the manuscript is not devoid of some major and minor problems, which require revision or better explanation to allow the reader understand the reliability of presented results. I see three major issues:
1) It is difficult to follow how the densities were estimated. The information to understand it step by step is available in the manuscript, but it is scattered in different places. For instance, to complete the information presented in the section of methods devoted to “density estimates” one has to look at the results and the Figure 3 to get the general idea. In the “density estimates” section a detailed approach is presented explaining that existing estimates from Zimbabwe and Kruger National Park were used to estimate the total regional cheetah population. This seems to suggest that these data were somehow used for the remaining part of the cheetah range, which is strange as the area of the cheetah presence in both Zimbabwe and Kruger is really minor relative to other populations (especially in Namibia). On the other hand, I learned from results that 14 empirical local cheetah population estimates were used, but it is unclear how all these data were combined with the Zimbabwe and Kruger data and analysed.
2) Although I appreciate the use of the Leslie matrix for estimation of the population growth, I do not understand how was it applied to estimate population densities, and specifically to account for the persecution data. Much information is provided both in the methods, results and the supplementary data referring to this analysis, however, it is not possible to follow the complete procedure and see the results of this analysis in the manuscript. I expect a graph resulting from this analysis showing the population trend based on the relationship between the growth rate and densities, both with and without persecution e.g. to illustrate the scenario mentioned in lines 370-372.
3) I also appreciate the authors’ efforts in trying to statistically support the “possible presence” area of the cheetah and estimating their potential numbers and densities. I understand that human related factors are generally negatively correlated with the cheetah presence. However, I also think that relying on this relationship directly is somewhat misused. There may be both, areas where predator still occurs with relatively high densities of livestock (actually seen on the Fig. 2 and areas where predator does not occur due to other than livestock/human issues. The authors seem to understand this problem as they acknowledge this in lines 445-448. Perhaps, accounting for another factor – wild prey occurrence – would add to the reliability of this estimation, but I assume such data may be equally difficult to get at sufficient resolution. Moreover, may be I missed this info, but I didn’t see the explanation how were these data on virtual cheetah presence farther stratified into densities. Anyway, considering the low reliability of the “possible” part of the cheetah range, I would put yet more emphasis on the untrustworthy character of this part of results.
Please note, I have included a number of other comments directly in the manuscript.

·

Basic reporting

no comment

Experimental design

no comment

Validity of the findings

no comment

---

## Round 0.2 · accepted · Accept

This is certainly an impressive paper. Having read the paper and the rebuttal letter I am happy with the corrections made and believe it significantly improved the manuscript. I am therefore happy to accept it for publication.